# Entropy-Calibrated Label Distribution Learning

**Yunan Lu**
Department of Computing
The Hong Kong Polytechnic University
Hong Kong, China
yunan.lu@polyu.edu.hk

**Bowen Xue**
Department of Computing
The Hong Kong Polytechnic University
Hong Kong, China
bowen.xue@connect.polyu.hk

**Xiuyi Jia**
School of Computer Science and Engineering
Nanjing University of Science and Technology
Nanjing, China
jiaxy@njust.edu.cn

**Lei Yang** [*]
Department of Computing
The Hong Kong Polytechnic University
Hong Kong, China
ray.yang@polyu.edu.hk

## Abstract

Label Distribution Learning (LDL) has emerged as a powerful framework for estimating complete conditional label distributions, providing crucial reliability for risk-sensitive decision-making tasks. While existing LDL algorithms exhibit competent performance under the conventional LDL performance evaluation methods, two key limitations remain: (1) current algorithms systematically underperform on the samples with low-entropy label distributions, which can be particularly valuable for decision making, and (2) the conventional performance evaluation methods are inherently biased due to the numerical imbalance of samples. In this paper, through empirical and theoretical analyses, we find that excessive cohesion between anchor vectors contributes significantly to the observed entropy bias phenomenon in LDL algorithms. Accordingly, we propose an inter-anchor angular regularization term that mitigates cohesion among anchor vectors by penalizing over-small angles. Besides, to alleviate the numerical imbalance of high-entropy samples in test set, we propose an entropy-calibrated aggregation strategy that obtains the overall model performance by evaluating performance on the low-entropy and high-entropy subsets of the overall test set separately. Finally, we conduct extensive experiments on various real-world datasets to demonstrate the effectiveness of our proposal.

## 1 Introduction

Accurately estimating the entire conditional distribution of labels (a.k.a. the label distribution) according to a set of feature variables, beyond merely the mean or mode of the distribution, is receiving increasing attention both in the field of statistics and machine learning [4, 28], as the information about the entire distribution is crucial in scenarios that are sensitive to risk, extremes, or uncertainty. To achieve this goal, researchers have developed various kinds of techniques, such as model calibration [27, 29] and mixture density neural network. These techniques aim to estimate the true label distributions using the training samples that are labeled only with the mean or mode of the underlying true label distribution, which are beneficial in practical tasks where the true label distributions are hardly available. However, there remain a large number of real-world scenarios where the true label distributions are readily available. For example, in rating prediction tasks or crowd-sourced learning, the labeling results for a given sample usually can be normalized as the

---

[*]Corresponding Author.

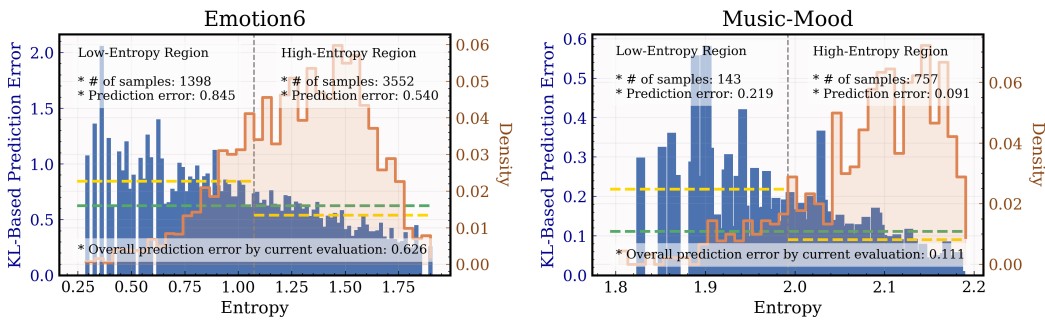

Figure 1: Distributions of the prediction error and the sample frequency w.r.t. the label distribution entropy on `Emotion6` [25] and `Music Mood` [13] datasets. Prediction error is measured by Kullback-Leibler (KL) divergence. Prediction error and sample density are denoted by blue and orange histograms, respectively. Each subfigure is partitioned into low-entropy and high-entropy regions by a vertical dashed line. The horizontal yellow dashed line in each region denotes the average prediction error of the test samples within that region. The horizontal green dashed line across the two regions denotes the overall prediction error of the test samples according to conventional evaluation methods.

proportion of participants who give each label or rating [2]; in drug efficacy prediction tasks where drug efficacy is quantified by the concentrations of a drug in the blood, the blood-drug concentration at different time points can be easily summarized as a drug concentration distribution by kernel density estimation [14]. The problem of learning samples with true label distributions is called Label Distribution Learning (LDL) [1]. Compared to the cases without true label distributions, LDL is capable of predicting the entire conditional distribution of labels more accurately, as it is directly supervised by the true label distributions.

Although existing LDL algorithms demonstrate strong performance under the conventional performance evaluation method, two critical issues emerge when delving into the distribution of prediction errors over test samples (visualized in Figure 1).

- First, current algorithms exhibit satisfactory prediction performance on high-entropy samples (i.e., the samples with high-entropy label distribution) yet markedly underperform on low-entropy samples (i.e., the samples with low-entropy label distribution), which can be demonstrated by the blue histograms in Figure 1. However, from a decision-theoretic perspective, low-entropy samples demand greater attention than high-entropy samples, as the former convey less uncertainty for practical decision-making.

- Second, conventional performance evaluation methods quantify the overall model performance by the arithmetic mean aggregation of performance metrics across all test samples. However, this approach inherently favors the numerical dominant samples. Since the high-entropy samples tends to outnumber the low-entropy samples in real-world tasks (as illustrated by the orange histograms in Figure 1), such aggregation typically fail to adequately capture the model performance on low-entropy samples.

Therefore, in this paper, we aim to address the entropy bias in current LDL algorithms and conventional model performance evaluation methods. Specifically, in terms of the LDL algorithm, we first analyze the generation mechanism of entropy bias from both empirical and theoretical perspectives, and consequently propose an assumption that the underperformance of LDL models on low-entropy samples is significantly driven by the cohesion of anchor vectors[2]. Based on these analyses, we propose IAR (i.e., an Inter-anchor Angular Regularization term) to penalize the anchor vectors with over-small angles. In terms of the performance evaluation method, we propose ECA (i.e., an Entropy-Calibrated Aggregation strategy) to calculate the overall model performance. Following the

---

[2]Typically, the output of LDL models can be expressed as $\mathrm{softmax}([\langle \boldsymbol{\omega}_m, \boldsymbol{v} \rangle]_{m=1}^M)$, where $\boldsymbol{v}$ denotes the feature vector of a sample. In the scenarios of deep learning, $\boldsymbol{v}$ is typically obtained by passing the raw feature vector $\boldsymbol{x}$ through a feature extraction network. In the scenarios of non-deep learning, $\boldsymbol{v}$ is usually set directly to the raw feature vector $\boldsymbol{x}$. Then, the set of vectors $\{\boldsymbol{\omega}_m\}_{m=1}^M$ constitutes the anchor vectors of the LDL model.

divide-and-conquer principle, ECA partitions the test set into low-entropy and high-entropy subsets based on a threshold. The average model performance is then computed separately for each subset. Finally, ECA evaluates the overall performance on the whole test set by the expected value of the average subset performance w.r.t. the threshold distribution. Empirically, extensive experiments on real-world datasets demonstrate that our proposal is effective in improving low-entropy sample predictions while maintaining satisfactory performance on high-entropy samples.

## 2 Related Work

Current research in LDL primarily focuses on two directions: loss function engineering and task-specific customization. The research on loss function enginerring mainly focuses on exploring either label correlations or sample correlations within label distributions. For example, LDLLC [5] constructs distance matrix from training label distributions to preserve label correlations during model learning. An algorithm based on optimal transport formulates the label correlation mining process as a metric learning problem, employing optimal transport distances to capture geometric relationships in the label space [37]. An algorithm based on local sample correlation introduces a local label correlation hypothesis, constructing sample-specific correlation vectors as additional features [38]. Differing from [38], an algorithm based on local low-rank label correlation [8] employs low-rank structures on local samples to discover label correlations. LCLR [26] simultaneously learns both global and local label correlations through low-rank approximation and clustering techniques. An algorithm based on label distribution manifold adopts a data-driven approach to leverage global and local correlations, learning the manifold structure of label distributions to constrain model outputs [30]. An algorithm based on fuzzy label correlation utilizes fuzzy membership-induced label correlation and joint fuzzy clustering and label correlation to capture multiple local label correlations [31]. TLRLDL [12] introduced an auxiliary multi-label learning process within the LDL framework, focusing on capturing low-rank label correlation within this auxiliary multi-label learning component rather than the LDL itself. Two algorithms based on label rankings propose to regularize the learning process by the label ranking correlation underlying the label distributions [6, 7]. Beyond fundamental loss function engineering, significant research efforts have been devoted to developing specialized LDL algorithms tailored for particular task requirements. For example, noise-robust LDL algorithms have been proposed to mitigate the adverse effects of inaccurate label distribution supervision during model training [3, 11, 21, 9]. LDL algorithms based on simple labels (e.g., binary labels [15, 18, 20, 33, 10], ternary labels [17], or label rankings [16, 19]) have been proposed to address the availability of label distribution supervision. LDL algorithms based on matrix completion have been proposed to learning the label distributions with missing values [32, 35, 36].

## 3 Methodology

This section presents our approach to addressing the entropy bias problem in label distribution learning. We begin by introducing the commonly-used mathematical notation and providing a problem formulation for LDL in Section 3.1. Building upon this foundation, Section 3.2 systematically investigates the underlying mechanisms responsible for the entropy bias. Finally, Section 3.3 elaborates on the technical details of our proposed loss function.

### 3.1 Problem Formulation

Let $\boldsymbol{x}$ and $\boldsymbol{y}$ denote the feature vector and the corresponding label distribution of a sample, respectively. Each element $y_m$ in $\boldsymbol{y}$ is called label description degree, and satisfies that $\sum_{m=1}^{M} y_m = 1$ and $y_m \geq 0$, where $M$ is the number of labels. The training set of LDL is denoted by $\{(\boldsymbol{x}_n, \boldsymbol{y}_n)\}_{n=1}^{N}$. LDL algorithms aim to learn a multivariate function $f : \boldsymbol{x} \mapsto \boldsymbol{y}$ based on the training set $\{(\boldsymbol{x}_n, \boldsymbol{y}_n)\}_{n=1}^{N}$.

### 3.2 Mechanisms of Entropy Bias in Label Distribution Learning Algorithms

Prior work by [19] preliminarily demonstrated that most baseline LDL algorithms exhibit a propensity for uniform label distributions. Inspired by this discovery, we present a more systematic visualization and analysis of the output entropy distribution across several recently proposed LDL algorithms [6, 7, 30, 31]. As shown in Figure 2, the prediction entropy (blue histogram) predominantly concentrates at

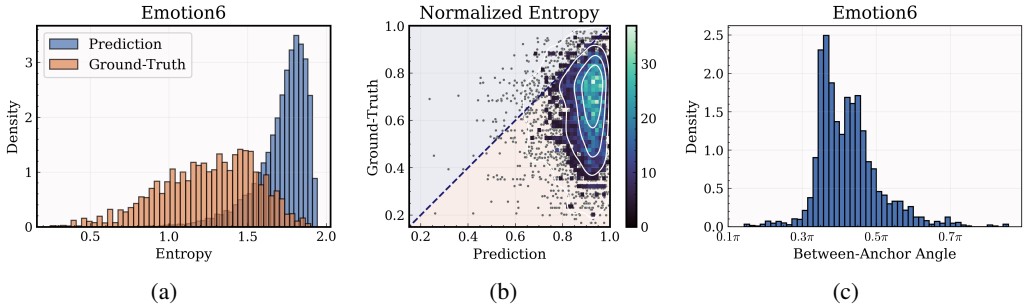

Figure 2: Entropy distributions of the prediction and the ground-truth label distributions. (a): The first subfigure depicts the entropy distributions of predicted versus ground-truth label distributions on `Emotion6` dataset. (b): The second subfigure presents their joint entropy distribution with sample-level visualization (gray points) and population density (kernel density plot). (c): The third subfigure depicts the distribution of the angles between anchors.

high entropy values, i.e., the prediction is over-uniform, whereas the ground-truth distribution (orange histogram) spans a broader range. This bias is further corroborated in the joint distribution, where sample density concentrates in the lower-right triangular region (semi-transparent red), confirming that the prediction entropy exceeds the ground-truth on most samples. Meanwhile, we visualize the distribution of the angles between the anchor vectors of the LDL models. It can be seen that the inter-anchor angle predominantly concentrates at low values, i.e., the anchors are cohesive. Intuitively, the cohesion of anchors and the over-uniformity of predictions typically occur in conjunction with each other. As illustrated in Figure 3(a), a strong correlation exists between smaller inter-anchor angles and reduced entropy values. Specifically, our visualization demonstrates an inverse relationship where narrower angular separation between anchors corresponds to lower entropy measures on average. This inverse relationship further suggests that the anchors with narrow angular separations are difficult to represent low-entropy samples. As illustrated in Figure 3(b), the space (indicated by the dark brown region) that can well-represent low-entropy samples is remarkably constrained when using anchors with narrow angular separation. In contrast, the anchors with wider angular separation exhibit substantially more expansive solution spaces for low-entropy sample representation. Furthermore, we propose Theorem 3.1 to rigorously verify the above idea, and the proof is provided in Appendix A.

**Theorem 3.1.** *Let $\{\boldsymbol{\omega}_m\}_{m=1}^{M}$ denote a group of cohesive anchor vectors, $\boldsymbol{x}$ denote the feature vector of a sample, and $\boldsymbol{y} = \mathrm{softmax}([\langle\boldsymbol{\omega}_m, \boldsymbol{x}\rangle]_{m=1}^{M})$ denote the corresponding output. Without loss of generality, we assume that the anchors are all unit vectors. Then Equation (1) holds if $\forall i \neq j, \angle(\boldsymbol{\omega}_i, \boldsymbol{\omega}_j) < \tau < \pi$, where $\angle(\boldsymbol{\omega}_i, \boldsymbol{\omega}_j)$ denotes the angle between anchors $\boldsymbol{\omega}_i$ and $\boldsymbol{\omega}_j$.*

$$\mathcal{H}(\boldsymbol{y}) \geq \frac{M\lambda^{\dagger} - 1}{\lambda^{\circ} - \lambda^{\dagger}} \cdot \lambda^{\circ} \log(\lambda^{\circ}) + \frac{M\lambda^{\circ} - 1}{\lambda^{\circ} - \lambda^{\dagger}} \lambda^{\dagger} \log(\lambda^{\dagger}) \tag{1}$$

*where $\lambda^{\dagger} = Z^{-1} \exp(\cos(\tau^{\circ} + \tau)\|\boldsymbol{x}\|)$, $\lambda^{\circ} = Z^{-1} \exp(\cos(\tau^{\circ})\|\boldsymbol{x}\|)$, $\mathcal{H}(\boldsymbol{y})$ denotes the entropy of the label distribution $\boldsymbol{y}$, $\tau^{\circ} = \min_m \angle(\boldsymbol{\omega}_m, \boldsymbol{x})$ is the minimum angle between $\boldsymbol{x}$ and anchors, $Z = \sum_{m=1}^{M} \exp(\langle\boldsymbol{\omega}_m, \boldsymbol{x}\rangle)$ denotes the normalization factor, and $\|\boldsymbol{x}\|$ denotes the $L_2$ norm of the feature vector $\boldsymbol{x}$. Equation (1) achieves equality when $M\lambda^{\circ} - 1 = k(\lambda^{\circ} - \lambda^{\dagger})$ and $k$ is a positive integer.*

Theorem 3.1 establishes the entropy range within which anchors with a certain degree ($\tau$) of cohesion can effectively represent samples. The derivative of the right-hand side of Equation (1) w.r.t. $\tau$ can be expressed as:

$$\frac{\lambda^{\circ}(M\lambda^{\circ} - 1)}{(\lambda^{\dagger} - \lambda^{\circ})^2} \cdot \left( \frac{\lambda^{\dagger}}{\lambda^{\circ}} - \log\left(\frac{\lambda^{\dagger}}{\lambda^{\circ}}\right) - 1 \right) \cdot \frac{\mathrm{d}\lambda^{\dagger}}{\mathrm{d}\tau}, \quad \frac{\mathrm{d}\lambda^{\dagger}}{\mathrm{d}\tau} = -\|\boldsymbol{x}\|(1 - \lambda^{\dagger})\lambda^{\dagger} \sin(\tau + \tau^{\circ}) \leq 0 \tag{2}$$

It is obvious that $(\lambda^{\dagger} - \lambda^{\circ})^{-2}\lambda^{\circ}(M\lambda^{\circ} - 1) \geq 0$. Besides, we have $\lambda^{\circ -1}\lambda^{\dagger} - \log(\lambda^{\circ -1}\lambda^{\dagger}) - 1 \geq 0$ according to the Taylor series expansion of the logarithmic function $\log(u)$ at $u = 1$. Therefore, the derivative of the right-hand side of Equation (1) w.r.t. $\tau$ is non-positive. In other words, decreasing $\tau$ causes the lower bound of $\mathcal{H}(\boldsymbol{y})$ to increase. Motivated by the above observations and analysis, we

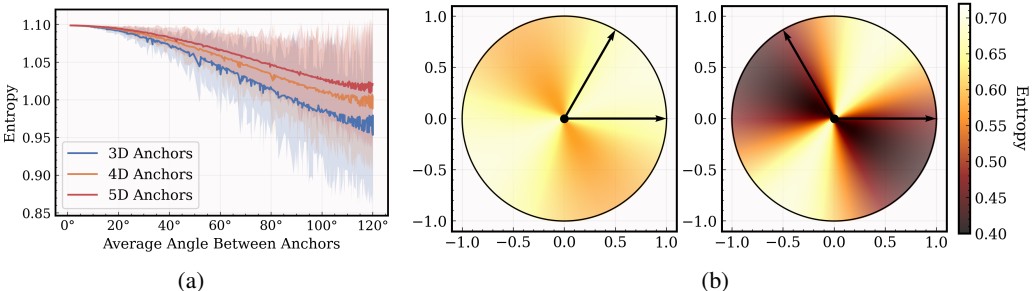

(a)                                                                    (b)

Figure 3: Relationship between the prediction entropy and the anchor angle. (a): The first subfigure shows the average prediction entropy (line) and the corresponding standard deviation (shadow) on different inter-anchor angles under one million random trivials. (b): The last two subfigures denote the entropy distribution under two 2D anchor vectors. The black directed lines denote the anchor vectors; the line between the zero point and a point on the unit circle represents a sample point whose color denotes the entropy of the predicted label distribution under the current anchor vectors.

argue that the underperformance of LDL models on low-entropy samples is significantly driven by the cohesion of anchors, which can be formalized as Assumption 3.2.

**Assumption 3.2.** Given two groups of anchor vectors $\mathcal{A}_1 = \{\boldsymbol{\omega}_m : \forall_{i \neq j} \angle(\boldsymbol{\omega}_i, \boldsymbol{\omega}_j) < \tau_1\}_{m=1}^M$ and $\mathcal{A}_2 = \{\boldsymbol{\omega}_m : \forall_{i \neq j} \angle(\boldsymbol{\omega}_i, \boldsymbol{\omega}_j) < \tau_2\}_{m=1}^M$, where $\angle(\boldsymbol{\omega}_i, \boldsymbol{\omega}_j)$ denotes the angle between the anchors $\boldsymbol{\omega}_i$ and $\boldsymbol{\omega}_j$, the prediction performance of the anchor vectors $\mathcal{A}_1$ on low-entropy samples tends to be inferior to that of the anchor vectors $\mathcal{A}_2$ if $\tau_1 < \tau_2$.

### 3.3 Inter-Anchor Angular Regularization

According to Assumption 3.2, we propose an inter-anchor angular regularization term (IAR) to penalize the small angles between anchors. The most straightforward formula for calculating the angle between anchors $\boldsymbol{\omega}_i$ and $\boldsymbol{\omega}_j$ is $\angle(\boldsymbol{\omega}_i, \boldsymbol{\omega}_j) = \arccos(\cos(\boldsymbol{\omega}_i, \boldsymbol{\omega}_j))$, which is not conducive to gradient calculations. Therefore, we minimize the cosine similarity between anchors $\boldsymbol{\omega}_i$ and $\boldsymbol{\omega}_j$, which is equivalent to maximizing their angular separation due to the monotonicity of the cosine function within the $[0, \pi]$ interval. Then the loss function can be formalized as Equation (3):

$$\mathcal{L} = \frac{1}{N} \sum_{n=1}^N \mathcal{D}_{\text{KL}}(f(\boldsymbol{x}_n) \| \boldsymbol{y}_n) + \frac{\alpha}{\#\mathbb{B}} \sum_{(i,j) \in \mathbb{B}} \frac{\langle \boldsymbol{\omega}_i, \boldsymbol{\omega}_j \rangle}{\|\boldsymbol{\omega}_i\| \|\boldsymbol{\omega}_j\|}, \tag{3}$$

where $\alpha$ is a trade-off hyperparameter, $\mathbb{B}$ is defined as $\{(i,j) : 1 \leq i < j \leq M\}$, $\#\mathbb{B}$ denotes the cardinality of $\mathbb{B}$, $\boldsymbol{\omega}_i$ denotes the anchor vectors, and $\mathcal{D}_{\text{KL}}(f(\boldsymbol{x}_n), \boldsymbol{y}_n)$ denotes the KL divergence between the ground-truth label distribution $\boldsymbol{y}_n$ and the model output $f(\boldsymbol{x}_n)$ of the sample $\boldsymbol{x}_n$, which is commonly utilized to encourage the model output to be closer to the ground-truth label distributions. The second term in Equation (3) is the inter-anchor angular regularization term, which penalize the small inter-anchor angle by minimizing the cosine similarity between anchors.

Next, we establish two mathematical properties of IAR to provide a more comprehensive understanding of its behavior in practical deployment. First, in order to facilitate optimization algorithms, we give the partial derivative of IAR w.r.t. anchor vectors:

$$\frac{\partial}{\partial \boldsymbol{\omega}_i} \cos(\boldsymbol{\omega}_i, \boldsymbol{\omega}_j) = \frac{1}{\|\boldsymbol{\omega}_i\|} \left( \frac{\boldsymbol{\omega}_j}{\|\boldsymbol{\omega}_j\|} - \cos(\boldsymbol{\omega}_i, \boldsymbol{\omega}_j) \frac{\boldsymbol{\omega}_i}{\|\boldsymbol{\omega}_i\|} \right). \tag{4}$$

Besides, we derive the value range of IAR. It is evident that IAR reaches its maximum value of $1$ when all anchor vectors are aligned in the same direction. However, the lower bound of IAR cannot reach $-1$ due to the geometric constraint that multiple vectors cannot be mutually antiparallel in all pairwise combinations. Therefore, we give the lower bound of IAR in Proposition 3.3.

**Proposition 3.3.** *Given any group of non-zero anchor vectors* $\{\boldsymbol{\omega}_m\}_{m=1}^{M}$*, Equation* (5) *holds for any* $M \geq 2$.

$$\frac{1}{\#\mathbb{B}} \sum_{(i,j)\in\mathbb{B}} \frac{\langle\boldsymbol{\omega}_i, \boldsymbol{\omega}_j\rangle}{\|\boldsymbol{\omega}_i\|\|\boldsymbol{\omega}_j\|} = \frac{1}{M(M-1)} \left( \left\| \sum_{m=1}^{M} \frac{\boldsymbol{\omega}_m}{\|\boldsymbol{\omega}_m\|} \right\|^2 - M \right) \geq -\frac{1}{M-1}. \quad (5)$$

Proposition 3.3 clearly demonstrates that the lower bound of IAR converges to 0 (i.e., the anchor vectors asymptotically approach mutual orthogonality) as the number of anchor vectors grows.

## 4 Performance Evaluation

In this section, we illustrate the proposed ECA (Entropy-Calibrated Aggregation) strategy to address the entropy bias in conventional model performance evaluation methods. Following the divide-and-conquer principle, the main idea underlying ECA is to evaluate the model performance separately on the low-entropy and high-entropy subsets of test set. Let $\mathcal{E}(f(\boldsymbol{x}_n), \boldsymbol{y}_n)$ denote the performance for the test sample $(\boldsymbol{x}_n, \boldsymbol{y}_n)$, $\mathcal{E}(\mathcal{C})$ denote the average performance for the set of test samples $\{(\boldsymbol{x}_n, \boldsymbol{y}_n)\}_{n\in\mathcal{C}}$. Conventional performance evaluation method compute the overall performance as:

$$\mathcal{E}(\mathcal{C}) = \frac{1}{\#\mathcal{C}} \sum_{n\in\mathcal{C}} \mathcal{E}(f(\boldsymbol{x}_n), \boldsymbol{y}_n) = \frac{\sum_{n\in\mathcal{C}_{\text{low}}} \mathcal{E}(f(\boldsymbol{x}_n), \boldsymbol{y}_n) + \sum_{n\in\mathcal{C}_{\text{high}}} \mathcal{E}(f(\boldsymbol{x}_n), \boldsymbol{y}_n)}{\#\mathcal{C}_{\text{low}} + \#\mathcal{C}_{\text{high}}}, \quad (6)$$

where $\mathcal{C} = \mathcal{C}_{\text{low}} \cup \mathcal{C}_{\text{high}}$ is the index set of test samples, $\mathcal{C}_{\text{low}}$ and $\mathcal{C}_{\text{high}}$ denote the index sets of low-entropy test samples and high-entropy test samples, respectively. It can be seen that if the number of high-entropy samples greatly exceeds the number of low-entropy samples (which is common in most practical situations), the conventional evauation method is heavily biased in favor of the high-entropy samples. and thus fails to adequately capture the model's performance on predicting low-entropy samples. To address this limitation, we can directly modify Equation (6) as Equation (7):

$$\frac{1}{2}(\mathcal{E}(\mathcal{C}_{\text{low}}) + \mathcal{E}(\mathcal{C}_{\text{high}})) = \frac{1}{2} \left( \frac{\sum_{n\in\mathcal{C}_{\text{low}}} \mathcal{E}(f(\boldsymbol{x}_n), \boldsymbol{y}_n)}{\#\mathcal{C}_{\text{low}}} + \frac{\sum_{n\in\mathcal{C}_{\text{high}}} \mathcal{E}(f(\boldsymbol{x}_n), \boldsymbol{y}_n)}{\#\mathcal{C}_{\text{high}}} \right) \quad (7)$$

Equation (7) first partitions the test set into low-entropy and high-entropy subsets, then computes the performance scores separately for each subset, and finally takes the average of the performance scores of the two subsets. Besides, we utilize a threshold to determine $\mathcal{C}_{\text{low}}$ and $\mathcal{C}_{\text{high}}$, i.e., $\mathcal{C}_{\text{low}}^{\kappa} = \{n \in \mathcal{C} : \mathcal{H}(\boldsymbol{y}_n) < \kappa\}$, $\mathcal{C}_{\text{high}}^{\kappa} = \{n \in \mathcal{C} : \mathcal{H}(\boldsymbol{y}_n) \geq \kappa\}$. The threshold $\kappa$ can be user-defined according to the specific task. For the sake of simplicity, we in this paper assume a uniform threshold $\kappa \sim \tilde{p}(\kappa)$, where $\tilde{p}(\kappa) = \texttt{Unif}(\kappa \mid \min_{n\in\mathcal{C}} \mathcal{H}(\boldsymbol{y}_n), \max_{n\in\mathcal{C}} \mathcal{H}(\boldsymbol{y}_n))$. Finally, we propose the evaluation method:

$$\frac{1}{2}\mathbb{E}_{\kappa\sim\tilde{p}(\kappa)} \left[ \mathcal{E}(\mathcal{C}_{\text{low}}^{\kappa}) + \mathcal{E}(\mathcal{C}_{\text{high}}^{\kappa}) \right] \approx \frac{1}{2T} \sum_{t=1}^{T} \mathcal{E}\left(\mathcal{C}_{\text{low}}^{\kappa^{(t)}}\right) + \mathcal{E}\left(\mathcal{C}_{\text{high}}^{\kappa^{(t)}}\right), \quad \kappa^{(t)} \sim \tilde{p}(\kappa), \quad (8)$$

where $T$ is the number of Monte Carlo samples utilized to approximate the intractable expectation.

## 5 Experiments

### 5.1 Experimental Configurations

**Datasets.** To ensure broad coverage of data complexity and practical scenarios, we select datasets including `Jaffe` ($\tilde{\mathcal{H}} : 0.96_{\pm0.03}$) [22], `BU-3DFE` ($\tilde{\mathcal{H}} : 0.95_{\pm0.04}$) [34], `Movie` ($\tilde{\mathcal{H}} : 0.88_{\pm0.06}$) [1], `Music Mood` ($\tilde{\mathcal{H}} : 0.94_{\pm0.03}$) [13], `Natural Scene` ($\tilde{\mathcal{H}} : 0.47_{\pm0.27}$) [1], `Emotion6` ($\tilde{\mathcal{H}} : 0.64_{\pm0.16}$) [25], `Art Painting` ($\tilde{\mathcal{H}} : 0.72_{\pm0.13}$) [23], and `M2B` ($\tilde{\mathcal{H}} : 0.41_{\pm0.12}$) [24], where $\tilde{\mathcal{H}}$ denotes the normalized entropy. More details are provided in Appendix B.1. Based on the entropy, the datasets can be categorized into the high-entropy group (from `Jaffe` to `Music Mood`) and the low-entropy group (from `Natural Scene` to `M2B`). Based on the task, the datasets cover emotion recognition (`Jaffe` and `BU-3DFE`), sentiment analysis (`Music Mood`, `Emotion6`, and `Art Painting`), scene recognition (`Natural Scene`), and rating prediction (`Movie` and `M2B`).

Table 1: Performance on High-Entropy Datasets (`Jaffe`, `BU-3DFE`, `Movie`, `Music Mood`).

| | KL (↓) | | | Cosine (↑) | | |
|---|---|---|---|---|---|---|
| | ECA | LEA | HEA | ECA | LEA | HEA |
| | | | `Jaffe` | | | |
| IAR | $\mathbf{0.041}_{\pm 0.005}$ | $\mathbf{0.053}_{\pm 0.019}$ | $0.044_{\pm 0.003}$ | $\mathbf{0.962}_{\pm 0.005}$ | $\mathbf{0.951}_{\pm 0.018}$ | $0.959_{\pm 0.003}$ |
| LDM | $\star 0.104_{\pm 0.015}$ | $\star 0.159_{\pm 0.029}$ | $\star 0.066_{\pm 0.008}$ | $\star 0.903_{\pm 0.015}$ | $\star 0.850_{\pm 0.029}$ | $\star 0.939_{\pm 0.006}$ |
| DPA | $\star 0.075_{\pm 0.010}$ | $\star 0.097_{\pm 0.025}$ | $\star 0.074_{\pm 0.006}$ | $\star 0.938_{\pm 0.008}$ | $\star 0.919_{\pm 0.024}$ | $\star 0.934_{\pm 0.005}$ |
| FCC | $\star 0.091_{\pm 0.015}$ | $\star 0.119_{\pm 0.040}$ | $\star 0.088_{\pm 0.008}$ | $\star 0.928_{\pm 0.011}$ | $\star 0.906_{\pm 0.033}$ | $\star 0.924_{\pm 0.006}$ |
| LRR | $\star 0.048_{\pm 0.005}$ | $\star 0.063_{\pm 0.018}$ | $\mathbf{0.043}_{\pm 0.002}$ | $\star 0.954_{\pm 0.004}$ | $\star 0.940_{\pm 0.017}$ | $\mathbf{0.961}_{\pm 0.002}$ |
| Ridge | $\star 0.093_{\pm 0.021}$ | $\star 0.133_{\pm 0.049}$ | $\star 0.067_{\pm 0.011}$ | $\star 0.916_{\pm 0.022}$ | $\star 0.876_{\pm 0.049}$ | $\star 0.939_{\pm 0.008}$ |
| | | | `BU-3DFE` | | | |
| IAR | $\mathbf{0.054}_{\pm 0.002}$ | $\mathbf{0.069}_{\pm 0.003}$ | $0.051_{\pm 0.002}$ | $\mathbf{0.948}_{\pm 0.002}$ | $\mathbf{0.936}_{\pm 0.003}$ | $0.949_{\pm 0.002}$ |
| LDM | $\star 0.119_{\pm 0.002}$ | $\star 0.190_{\pm 0.003}$ | $\star 0.056_{\pm 0.002}$ | $\star 0.888_{\pm 0.002}$ | $\star 0.822_{\pm 0.003}$ | $\star 0.945_{\pm 0.002}$ |
| DPA | $\star 0.057_{\pm 0.003}$ | $\star 0.072_{\pm 0.004}$ | $0.051_{\pm 0.002}$ | $\star 0.945_{\pm 0.003}$ | $\star 0.932_{\pm 0.004}$ | $0.949_{\pm 0.002}$ |
| FCC | $\star 0.058_{\pm 0.003}$ | $\star 0.072_{\pm 0.004}$ | $\star 0.055_{\pm 0.003}$ | $\star 0.945_{\pm 0.003}$ | $\star 0.934_{\pm 0.004}$ | $\star 0.946_{\pm 0.003}$ |
| LRR | $\star 0.057_{\pm 0.002}$ | $\star 0.075_{\pm 0.003}$ | $\mathbf{0.049}_{\pm 0.002}$ | $\star 0.945_{\pm 0.002}$ | $\star 0.929_{\pm 0.003}$ | $\mathbf{0.951}_{\pm 0.002}$ |
| Ridge | $\star 0.110_{\pm 0.002}$ | $\star 0.171_{\pm 0.003}$ | $\star 0.055_{\pm 0.002}$ | $\star 0.895_{\pm 0.002}$ | $\star 0.837_{\pm 0.002}$ | $\star 0.945_{\pm 0.002}$ |
| | | | `Movie` | | | |
| IAR | $\mathbf{0.259}_{\pm 0.055}$ | $\mathbf{0.437}_{\pm 0.119}$ | $0.097_{\pm 0.002}$ | $\mathbf{0.852}_{\pm 0.025}$ | $\mathbf{0.747}_{\pm 0.064}$ | $0.935_{\pm 0.002}$ |
| LDM | $\star 0.326_{\pm 0.032}$ | $\star 0.508_{\pm 0.078}$ | $\star 0.180_{\pm 0.004}$ | $\star 0.805_{\pm 0.013}$ | $\star 0.710_{\pm 0.038}$ | $\star 0.870_{\pm 0.002}$ |
| DPA | $\star 0.262_{\pm 0.056}$ | $\star 0.446_{\pm 0.128}$ | $\star 0.103_{\pm 0.006}$ | $\star 0.849_{\pm 0.025}$ | $\star 0.742_{\pm 0.066}$ | $\star 0.932_{\pm 0.003}$ |
| FCC | $\star 0.284_{\pm 0.058}$ | $\star 0.481_{\pm 0.129}$ | $\star 0.134_{\pm 0.005}$ | $\star 0.841_{\pm 0.026}$ | $\star 0.730_{\pm 0.063}$ | $\star 0.916_{\pm 0.003}$ |
| LRR | $\star 0.262_{\pm 0.054}$ | $\star 0.442_{\pm 0.122}$ | $\circ \mathbf{0.090}_{\pm 0.003}$ | $\star 0.850_{\pm 0.024}$ | $\star 0.743_{\pm 0.067}$ | $\circ \mathbf{0.940}_{\pm 0.002}$ |
| Ridge | $\star 0.338_{\pm 0.048}$ | $\star 0.593_{\pm 0.143}$ | $\star 0.109_{\pm 0.005}$ | $\star 0.807_{\pm 0.020}$ | $\star 0.654_{\pm 0.077}$ | $\star 0.928_{\pm 0.003}$ |
| | | | `Music Mood` | | | |
| IAR | $\mathbf{0.136}_{\pm 0.013}$ | $\mathbf{0.188}_{\pm 0.018}$ | $\mathbf{0.082}_{\pm 0.005}$ | $\mathbf{0.905}_{\pm 0.008}$ | $\mathbf{0.875}_{\pm 0.010}$ | $\mathbf{0.933}_{\pm 0.004}$ |
| LDM | $\star 0.154_{\pm 0.013}$ | $\star 0.215_{\pm 0.029}$ | $\star 0.094_{\pm 0.005}$ | $\star 0.891_{\pm 0.008}$ | $\star 0.854_{\pm 0.019}$ | $\star 0.923_{\pm 0.005}$ |
| DPA | $\star 0.146_{\pm 0.015}$ | $\star 0.201_{\pm 0.023}$ | $\star 0.087_{\pm 0.007}$ | $\star 0.898_{\pm 0.009}$ | $\star 0.865_{\pm 0.014}$ | $\star 0.930_{\pm 0.006}$ |
| FCC | $\star 0.156_{\pm 0.014}$ | $\star 0.205_{\pm 0.023}$ | $\star 0.099_{\pm 0.009}$ | $\star 0.890_{\pm 0.010}$ | $\star 0.860_{\pm 0.014}$ | $\star 0.920_{\pm 0.007}$ |
| LRR | $\star 0.143_{\pm 0.014}$ | $\star 0.197_{\pm 0.022}$ | $0.083_{\pm 0.007}$ | $\star 0.900_{\pm 0.008}$ | $\star 0.867_{\pm 0.014}$ | $0.932_{\pm 0.006}$ |
| Ridge | $\star 0.150_{\pm 0.016}$ | $\star 0.202_{\pm 0.024}$ | $\star 0.089_{\pm 0.008}$ | $\star 0.895_{\pm 0.010}$ | $\star 0.863_{\pm 0.016}$ | $\star 0.928_{\pm 0.007}$ |

**Evaluation Measures.** Considering the suggestion proposed in [1] and the page limit, we employ both KL divergence and cosine similarity as evaluation metrics for individual sample. The better performance is represented by the higher value of KL divergence (↑) or the lower value of cosine similarity (↓). For the overall performance assessment, we implement three aggregation approaches. The first one is our proposed ECA method (presented in Section 4), where $T$ is set to 10. The second one is LEA (Low Entropy Aggregation), which only computes the average performance across low-entropy test samples. The third one is HEA (High Entropy Aggregation), which only computes the average performance across high-entropy test samples. The entropy threshold for sample partition is defined as the arithmetic mean of the maximum and minimum entropy values within the test set.

## 5.2 Comparison Algorithms and Experimental Procedure

**Comparison Algorithms.** We employ four recently proposed LDL algorithms for comparative study, including LDM [30], DPA [6], FCC [31], and LRR [7]. All hyperparameters for these comparison algorithms are tuned within the ranges recommended by their respective publications. For our proposed method, the hyperparameter $\alpha$ is optimized within the range of $\{1, 10, 20, \dots, 100\}$. We employ L-BFGS to minimize the loss function of our method. Furthermore, to ensure fair comparison, we set the trade-off parameters of the $L_2$ regularization in comparison algorithms as 0, consistent with the implementation of all comparison algorithms. To compare our proposed

Table 2: Performance on Low-Entropy Datasets (`Natural Scene`, `Emotion6`, `Art Painting`, `M2B`).

| | KL ($\downarrow$) | | | Cosine ($\uparrow$) | | |
|---|---|---|---|---|---|---|
| | ECA | LEA | HEA | ECA | LEA | HEA |
| | | | `Natural Scene` | | | |
| IAR | $0.736_{\pm0.033}$ | $\mathbf{0.892}_{\pm0.037}$ | $0.655_{\pm0.033}$ | $0.760_{\pm0.009}$ | $\mathbf{0.756}_{\pm0.010}$ | $0.746_{\pm0.010}$ |
| LDM | $\star 1.237_{\pm0.007}$ | $\star 1.901_{\pm0.014}$ | $\star 0.786_{\pm0.008}$ | $\star 0.567_{\pm0.002}$ | $\star 0.388_{\pm0.003}$ | $\star 0.675_{\pm0.003}$ |
| DPA | $\star 0.774_{\pm0.041}$ | $\star 0.907_{\pm0.041}$ | $\star 0.699_{\pm0.034}$ | $\star 0.750_{\pm0.008}$ | $\star 0.751_{\pm0.011}$ | $\star 0.733_{\pm0.010}$ |
| FCC | $\star 1.049_{\pm0.080}$ | $\star 1.057_{\pm0.054}$ | $\star 0.999_{\pm0.069}$ | $\star 0.698_{\pm0.009}$ | $\star 0.715_{\pm0.011}$ | $\star 0.670_{\pm0.010}$ |
| LRR | $\circ \mathbf{0.709}_{\pm0.018}$ | $\star \mathit{0.906}_{\pm0.031}$ | $\circ \mathbf{0.612}_{\pm0.012}$ | $\circ \mathbf{0.768}_{\pm0.006}$ | $\star \mathit{0.753}_{\pm0.011}$ | $\circ \mathbf{0.761}_{\pm0.004}$ |
| Ridge | $\star 0.980_{\pm0.008}$ | $\star 1.406_{\pm0.013}$ | $\star 0.715_{\pm0.015}$ | $\star 0.663_{\pm0.005}$ | $\star 0.566_{\pm0.004}$ | $\star 0.710_{\pm0.009}$ |
| | | | `Emotion6` | | | |
| IAR | $0.689_{\pm0.054}$ | $\mathbf{0.810}_{\pm0.035}$ | $0.490_{\pm0.027}$ | $0.693_{\pm0.021}$ | $\mathbf{0.667}_{\pm0.015}$ | $0.741_{\pm0.011}$ |
| LDM | $\star 0.801_{\pm0.022}$ | $\star 1.078_{\pm0.028}$ | $\star 0.541_{\pm0.021}$ | $\star 0.639_{\pm0.008}$ | $\star 0.534_{\pm0.009}$ | $\star 0.710_{\pm0.008}$ |
| DPA | $\star 0.709_{\pm0.054}$ | $\star 0.816_{\pm0.039}$ | $\star 0.511_{\pm0.019}$ | $\star 0.684_{\pm0.020}$ | $\star 0.661_{\pm0.016}$ | $\star 0.733_{\pm0.007}$ |
| FCC | $\star 0.717_{\pm0.055}$ | $\star 0.817_{\pm0.035}$ | $\star 0.528_{\pm0.019}$ | $\star 0.680_{\pm0.019}$ | $\star 0.661_{\pm0.012}$ | $\star 0.726_{\pm0.007}$ |
| LRR | $\circ \mathbf{0.675}_{\pm0.053}$ | $\star \mathit{0.815}_{\pm0.036}$ | $\circ \mathbf{0.472}_{\pm0.018}$ | $\circ \mathbf{0.699}_{\pm0.022}$ | $\star \mathit{0.661}_{\pm0.016}$ | $\circ \mathbf{0.749}_{\pm0.008}$ |
| Ridge | $\star 0.711_{\pm0.021}$ | $\star 0.900_{\pm0.047}$ | $\star 0.496_{\pm0.015}$ | $\star 0.677_{\pm0.009}$ | $\star 0.620_{\pm0.019}$ | $\star 0.733_{\pm0.007}$ |
| | | | `Art Painting` | | | |
| IAR | $0.650_{\pm0.128}$ | $\mathbf{0.777}_{\pm0.092}$ | $0.498_{\pm0.046}$ | $0.709_{\pm0.046}$ | $\mathbf{0.687}_{\pm0.026}$ | $0.740_{\pm0.018}$ |
| LDM | $0.869_{\pm0.517}$ | $\star 1.037_{\pm0.273}$ | $0.572_{\pm0.253}$ | $\star 0.657_{\pm0.061}$ | $\star 0.574_{\pm0.045}$ | $0.723_{\pm0.040}$ |
| DPA | $\star 0.906_{\pm0.207}$ | $\star 0.965_{\pm0.177}$ | $\star 0.695_{\pm0.137}$ | $\star 0.645_{\pm0.058}$ | $\star 0.657_{\pm0.037}$ | $\star 0.691_{\pm0.025}$ |
| FCC | $\star 1.186_{\pm0.285}$ | $\star 1.193_{\pm0.357}$ | $\star 0.970_{\pm0.200}$ | $\star 0.603_{\pm0.053}$ | $\star 0.624_{\pm0.043}$ | $\star 0.648_{\pm0.034}$ |
| LRR | $\mathbf{0.646}_{\pm0.128}$ | $\star \mathit{0.807}_{\pm0.120}$ | $\circ \mathbf{0.462}_{\pm0.040}$ | $0.713_{\pm0.042}$ | $\star \mathit{0.671}_{\pm0.025}$ | $\circ \mathbf{0.755}_{\pm0.015}$ |
| Ridge | $\star 0.724_{\pm0.124}$ | $\star 0.923_{\pm0.150}$ | $\star 0.506_{\pm0.066}$ | $\star 0.677_{\pm0.035}$ | $\star 0.617_{\pm0.050}$ | $0.734_{\pm0.019}$ |
| | | | `M2B` | | | |
| IAR | $\mathbf{0.744}_{\pm0.074}$ | $\mathbf{0.906}_{\pm0.043}$ | $\mathbf{0.321}_{\pm0.018}$ | $\mathbf{0.713}_{\pm0.016}$ | $\mathbf{0.608}_{\pm0.016}$ | $\mathbf{0.838}_{\pm0.013}$ |
| LDM | $\star 0.956_{\pm0.056}$ | $\star 1.150_{\pm0.028}$ | $\star 0.801_{\pm0.064}$ | $\star 0.591_{\pm0.032}$ | $\star 0.516_{\pm0.016}$ | $\star 0.635_{\pm0.038}$ |
| DPA | $\star 1.022_{\pm0.139}$ | $\star 1.252_{\pm0.108}$ | $\star 0.502_{\pm0.053}$ | $\star 0.667_{\pm0.026}$ | $\star 0.570_{\pm0.025}$ | $\star 0.773_{\pm0.017}$ |
| FCC | $\star 1.081_{\pm0.163}$ | $\star 1.311_{\pm0.087}$ | $\star 0.568_{\pm0.055}$ | $\star 0.661_{\pm0.029}$ | $\star 0.564_{\pm0.021}$ | $\star 0.759_{\pm0.017}$ |
| LRR | $\star \mathit{0.773}_{\pm0.089}$ | $\star \mathit{0.946}_{\pm0.042}$ | $\star \mathit{0.338}_{\pm0.018}$ | $\star \mathit{0.704}_{\pm0.019}$ | $\star \mathit{0.601}_{\pm0.017}$ | $\star \mathit{0.824}_{\pm0.010}$ |
| Ridge | $\star 0.816_{\pm0.073}$ | $\star 1.011_{\pm0.120}$ | $\star 0.377_{\pm0.078}$ | $\star 0.696_{\pm0.014}$ | $\star 0.596_{\pm0.022}$ | $\star 0.814_{\pm0.022}$ |

IAR term and $L_2$ regularization term, we introduce ridge regression for LDL, which minimizes $\frac{1}{N}\sum_{n=1}^{N}\mathcal{D}_{\mathrm{KL}}(f(\boldsymbol{x}_n)\|\boldsymbol{y}_n) + \frac{\alpha}{M}\sum_{m=1}^{M}\|\boldsymbol{\omega}_m\|$, and $\alpha$ is selected from $\{10^{-3}, 10^{-2}, \ldots, 10^3\}$. Besides, we normalize the feature data to improve the convergence stability of all comparison algorithms.

**Experimental Procedure.** Given a dataset with label distributions, we first randomly divide the dataset into two subsets (30% is used as the test set and 70% is used as the training set). Further, we train an LDL model on the training set and apply the model to predict the label distribution of the test samples. Then, we evaluate the performance of the LDL model by comparing the ground-truth and the predicted label distributions. Finally, we repeat the above process ten times under randomly different dataset partitions and statistically summarize the results of the ten random experiments.

### 5.3 Discussion on Experimental Results

As shown in Tables 1 and 2, the results are interpreted as follows: each cell entry (e.g., $\star 0.104_{\pm0.015}$) indicates the mean performance ($\pm$ standard deviation); the symbol "$\star$" denotes the cases where our proposed IAR is statistically superior to the corresponding algorithm under paired two-tailed $t$-test with $p < 0.05$, while "$\circ$" denotes the significant inferiority of IAR. Absence of annotations implies no statistically significant difference. Boldface and italics highlight the best and second best performance within each comparison group. The experimental results show that IAR performs outstandingly on the

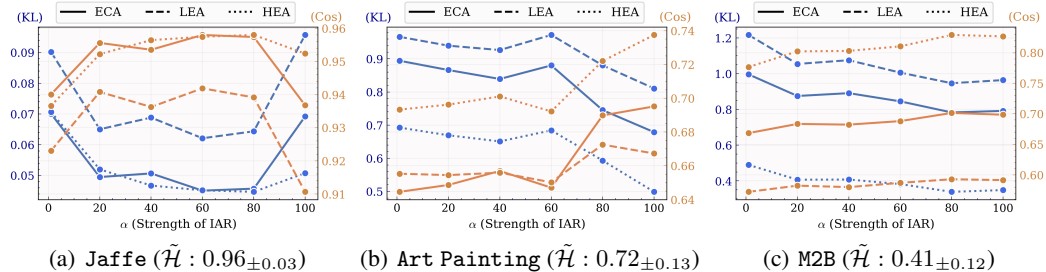

(a) Jaffe ($\tilde{\mathcal{H}} : 0.96_{\pm 0.03}$)    (b) Art Painting ($\tilde{\mathcal{H}} : 0.72_{\pm 0.13}$)    (c) M2B ($\tilde{\mathcal{H}} : 0.41_{\pm 0.12}$)

Figure 4: Prediction performance on varying $\alpha$. The performance quantified by KL divergence (KL) and cosine similarity (Cos) is represented by blue and red, respectively. The performance evaluated by ECA, LEA, and HEA is represented by the solid lines, dashed lines, and dotted lines, respectively.

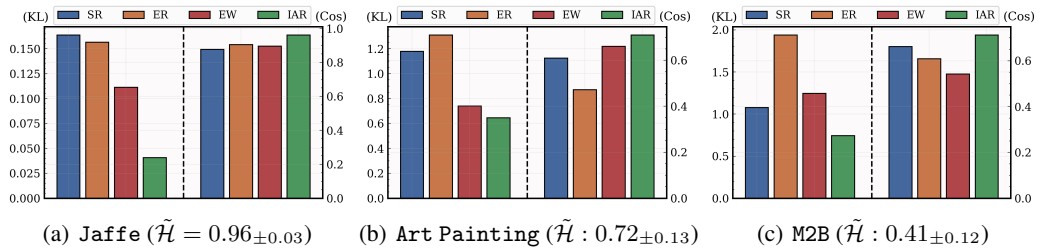

(a) Jaffe ($\tilde{\mathcal{H}} = 0.96_{\pm 0.03}$)    (b) Art Painting ($\tilde{\mathcal{H}} : 0.72_{\pm 0.13}$)    (c) M2B ($\tilde{\mathcal{H}} : 0.41_{\pm 0.12}$)

Figure 5: Prediction performance of ablation algorithms evaluated by ECA. Each subfigure is bisected by a vertical dashed line, with the left and right sides representing the performance measured by KL divergence (KL) and cosine similarity (Cos), respectively.

high-entropy datasets (Table 1), while it performs sub-optimally on the low-entropy datasets (Table 2). In terms of the high-entropy datasets, our IAR demonstrates statistically significant superiority over all competitors under both ECA and LEA evaluation method. When evaluated by HEA, IAR either achieves top performance or shows no statistically significant difference from the top-performing competitor. The sole exception occurs on the Movie dataset, where IAR is significantly outperformed by LRR. In terms of the low-entropy datasets, while IAR still significantly outperforms all competitors on low-entropy samples, it sacrifices prediction performance for high-entropy samples compared to LRR on Natural Scene, Emotion6, and Art Painting. This aligns with our expectations. Since the model trained by high-entropy samples is more likely to output the over-squeezed anchor vectors, which can be effectively avoided by adding IAR term. On the contrary, in order to fit the low-entropy label distributions, the model trained by low-entropy samples is not prone to output the over-squeezed anchor vectors, and thus the model cannot benefit significantly from IAR term. Nonetheless, IAR consistently performs best on low-entropy samples, which suggests that IAR is able to improve the prediction performance for low-entropy samples to varying degrees on various real-world datasets. More experiments demonstrating the effectiveness of IAR can be found in Appendix B.2.

### 5.4 Further Analysis

**Hyperparameter Sensitivity.** Figure 4 presents the impact of hyperparameter $\alpha$ on the KL metric under the ECA evaluation method, demonstrating the performance sensitivity of our method w.r.t. the hyperparameter variations. Experimental results demonstrate that moderately increasing the IAR weight (e.g., $\alpha \in [1, 40]$) consistently enhances model performance. However, excessive values (e.g., $\alpha > 40$) should be applied with caution, as excessive $\alpha$ may induce severe underfitting accompanied by significant performance degradation on both low-entropy and high-entropy samples. Furthermore, for the datasets that most samples possess the label distribution with low entropy (e.g., "M2B" dataset), we can safely employ relatively large $\alpha$ values for better model performance.

**Ablation Study.** We construct three ablation variants for comprehensive analysis. The first one is the LDLIAR with $\alpha$ zeroed out, i.e., a simple softmax regression, which is abbreviated as "SR".

The second one is an LDL algorithm with entropy regression, which is abbreviated as "ER", whose loss function for each sample is defined by $\mathcal{D}_{\mathrm{KL}}(f(\boldsymbol{x}_n)\|\boldsymbol{y}_n) + \lambda|\mathcal{H}(f(\boldsymbol{x}_n)) - \mathcal{H}(\boldsymbol{y}_n)|$. The third one is an entropy-weighted LDL algorithm, which is abbreviated as "EW", whose loss function for each sample is defined by $\exp(-\lambda \cdot \mathcal{H}(\boldsymbol{y}_n)) \cdot \mathcal{D}_{\mathrm{KL}}(f(\boldsymbol{x}_n)\|\boldsymbol{y}_n)$. To address entropy bias, "ER" incorporates an additional loss term that penalizes the entropy discrepancy between the predictions and the ground-truth, while "EW" assigns higher weights to the low-entropy samples to prioritize their contribution during model training. "ER" and "EW" implement two straightforward approaches for addressing entropy bias. Their hyperparameter $\lambda$ is selected from $\{10^{-3}, 10^{-2}, \ldots, 10^{3}\}$. The experimental results are shown in Figure 5, which demonstrates that our proposed IAR achieves significant improvements compared to other ablation algorithms.

## 6 Limitations and Conclusion

**Limitations.** First, our proposed inter-anchor angular regularization (IAR) term is not directly compatible with tree-based LDL algorithms, as their learning process do not involve anchor vectors. Second, Theorem 3.1 assumes an output layer with non-negative gradients (e.g., softmax normalization). Consequently, the theoretical guarantees do not hold for the output layers with activation or normalization functions that violate the non-negativity.

**Conclusion.** In this work, we systematically investigate the limitations of existing algorithms and evaluation methods on label distribution learning in handling low-entropy samples. In terms of the algorithm, we reveal, from both empirical and theoretical perspectives, that excessive cohesion of anchor vectors is an essential cause of the underperformance on low-entropy samples. According to this assumption, we propose an inter-anchor angular regularization (IAR) term to explicitly penalizes over-squeezed angular separations between anchor vectors, and derive several mathematical properties that can be beneficial for practical use. In terms of the performance evaluation method, we introduce an entropy-calibrated aggregation (ECA) method to avoid the imbalance between low-entropy and high-entropy samples. Finally, extensive experimental results verify the validity of our proposal.

## 7 Acknowledgements

This work was partially supported by the National Natural Science Foundation of China (62176123, 62476130), the Natural Science Foundation of Jiangsu Province (BK20242045), the Innovation and Technology Fund (GHP/079/22SZ, ITS/034/23FP), and the UGC/GRF (No. 15211024, 15215421).

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
