# OpenReview forum: "Entropy-Calibrated Label Distribution Learning"
_NeurIPS.cc/2025/Conference — NeurIPS 2025 poster_

### Official Review · Reviewer_Z4dW · 2025-06-17

**Clarity:** 3
**Significance:** 3
**Originality:** 3
**Rating:** 5
**Confidence:** 4

**Summary:**

This paper points out that in the current field of label distribution learning, the existing algorithms underperform on low-entropy samples, and the existing evaluation metrics of label distribution learning cannot adequately capture the prediction performance of samples with different levels of entropy. Aiming at these two problems, this paper theoretically analyzes the underlying reasons for the underperformance on low-entropy samples, designs a regularization term according to the theoretical results, and proposes a novel evaluation method for label distribution learning.

**Questions:**

1. In order to make the anchors as dispersed as possible, why not just minimize the inner product of the anchors instead of minimizing the cosine similarity of the two anchors? After all, cosine similarity imposes a greater computational burden compared to inner product?
2. Are there some empirical suggestions for hyperparameter selection?

**Ethical Concerns:**

["NO or VERY MINOR ethics concerns only"]

**Final Justification:**

The authors have addressed my concerns and thus I maintain my previous rating.

**Limitations:**

yes

**Quality:**

3

**Strengths And Weaknesses:**

Strengths: 1) This paper presents the problem of underperformance on low-entropy samples and the bias of evaluation metrics in existing LDL studies, which are widespread and have serious impacts on the downstream decision-making tasks of LDL because low-entropy samples usually harbor expensive misjudgement costs. 2) This paper analyzes the causes of the underperformance problem of existing LDL algorithms on low-entropy samples from the theoretical and empirical viewpoints. The theoretical and visualization results reliably reveal the relationship between the hypothesis space and the low-entropy underperformance of the widely-adopted softmax model, which is instructive for the design of subsequent LDL algorithms. 3) This paper proposes a regularization term based on the results of the theoretical analysis, gives several mathematical properties of the regularization term, and verifies the validity of the regularization term through extensive experiments. 4) This paper proposes a new evaluation method that can effectively capture the prediction performance of samples with different levels of entropy.

Weaknesses: 1) The experimental configurations of this paper are not adequately shown, such as the specific configuration of the optimizer used and the iteration stopping conditions. 2) The writing of the paper could be further improved, such as the mismatch of left and right parentheses in line 196, the lack of period at the end of the sentence in line 197, and in the captions of the subfigures in Fig. 5, Fig. (b) and Fig. (c) show the entropy mean and the entropy variance of the dataset but Fig. (a) shows only the mean value. The paragraph subheadings before line 267 end with a period, while the paragraph subheadings after line 267 end without a period, which is not uniform in format. The dataset name in the image caption in Figure 1 does not match the dataset name in the caption.

---

> ### Author Rebuttal · Authors · 2025-07-30
>
> Dear Reviewer Z4dW,
>
> Thank you for your positive assessment on our work. We sincerely appreciate the time and effort that you have dedicated to evaluating our work. We have carefully considered all the comments and have revised the paper accordingly. Below, we provide a point-by-point response to your suggestions.
>
> ***Weakness 1: The experimental configurations of this paper are not adequately shown, such as the specific configuration of the optimizer used and the iteration stopping conditions.***
>
> We have added more necessary information about the experimental configurations. The optimizer used in our paper is the L-BFGS algorithm implemented by pytorch. The code is shown as follows:
>
> ```python
> optimizer = torch.optim.LBFGS(..., lr=1e-3, max_iter=1000, tolerance_grad=1e-5, tolerance_change=1.4901161193847656e-8, history_size=5, line_search_fn='strong_wolfe', max_eval=None)
> ```
>
> The above code shows that the optimization process will be stopped when the change of loss function is smaller than 1.4901161193847656e-8 or the gradient is smaller than 1e-5.
>
> ***Weakness 2: The writing of the paper could be further improved.***
>
> We have improved the writing of our paper. For example, we have added a left parentheses before the "Art Painting"; we have added a period at the end of the sentence in line 197; we have added the entropy variance in the caption of Fig. 5(a); we have calibrated the inconsistency of period of the paragraph subheadings.
>
> ***Question 1: In order to make the anchors as dispersed as possible, why not just minimize the inner product of the anchors instead of minimizing the cosine similarity of the two anchors?***
>
> The inner product of anchor vectors is also determined by their norm, which undermines its reliability in preventing over-uniformity. Mathematically, the inner product between anchor vectors equals to their cosine similarity multiplied by their $L_2$ norms. Consequently, the minimization of the inner product could result merely from reduced vector norms rather than improved angular dispersion (i.e., decreased cohesion among anchor vectors).
>
> ***Question 2: Are there some empirical suggestions for hyperparameter selection?***
>
> As analyzed in lines 249-254 of our paper, we recommend $\alpha=40$ as a default setting for most datasets, as it approaches optimal performance across diverse scenarios. For further performance optimization, we suggest the following empirical guidelines:
>
> 1. When the training set contains predominantly high-entropy label distributions, consider using $\alpha <40$.
> 2. For datasets with predominantly low-entropy label distributions, larger values are preferable.

---

### Official Review · Reviewer_zUMv · 2025-06-22

**Clarity:** 2
**Significance:** 3
**Originality:** 3
**Rating:** 4
**Confidence:** 3

**Summary:**

This paper addresses the problem of entropy bias in Label Distribution Learning (LDL), where models tend to perform poorly on low-entropy samples due to overly cohesive anchor vectors. To mitigate this, the authors propose an Inter-anchor Angular Regularization (IAR) term that penalizes anchor vectors with small angular separation, thereby enhancing the model’s ability to represent low-entropy samples. Additionally, they introduce an Entropy-Calibrated Aggregation (ECA) strategy for fairer model evaluation by separately assessing performance on low- and high-entropy subsets. Experimental results demonstrate the effectiveness of the proposed methods.

**Questions:**

1.	What is the form of the function $f (\cdot)$? Does it use only anchor vectors as classifier weights?

2.	Are the anchor vectors treated as learnable parameters? It would be helpful to clarify the background of LDL.

**Ethical Concerns:**

["NO or VERY MINOR ethics concerns only"]

**Final Justification:**

All of my concerns have been addressed, and I am inclined to maintain a positive score for this work.

**Limitations:**

The limitations are included in Section 6.

**Paper Formatting Concerns:**

No.

**Quality:**

3

**Strengths And Weaknesses:**

Strengths

1.	The paper tackles the entropy bias in LDL, which is interesting.

2.	The proposed method is built upon both empirical and theoretical analyses, and is well-motivated.

3.	This work also proposes a new evaluation mechanism termed ECA for fairer model evaluation, which sounds reasonable.

Weaknesses

1.	In line 42, “the samples with high-entropy” should be “the samples with low-entropy”.

2.	For better clarity, it would be beneficial to mention the term “ECA” in Section 4.

3.	In the main text, it would be beneficial to reference the Appendices explicitly. For example, after presenting Theorem 3.1, a note should be added indicating that its proof is provided in Appendix A.

---

> ### Author Rebuttal · Authors · 2025-07-30
>
> Dear Reviewer zUMv,
>
> Thank you for your positive assessment on our work. We sincerely appreciate the time and effort that you have dedicated to evaluating our work. We have carefully considered all the comments and have revised the paper accordingly. Below, we provide a point-by-point response to your suggestions.
>
> ***Weakness 1: In line 42, "the samples with high-entropy" should be "the samples with low-entropy".***
>
> As suggested, we have corrected this typo error in the revised version.
>
> ***Weakness 2: For better clarity, it would be beneficial to mention the term "ECA" in Section 4.***
>
> As suggested, we have added an introductory paragraph about ECA at the beginning of Section 4 to elucidate its objectives and main concepts. The details are as follows. In this section, we illustrate the proposed ECA (Entropy-Calibrated Aggregation) strategy to address the entropy bias in conventional model performance evaluation methods. Following the divide-and-conquer principle, the main idea underlying ECA is to evaluate the model performance separately on the low-entropy and high-entropy subsets of test set.
>
> ***Weakness 3: In the main text, it would be beneficial to reference the Appendices explicitly. For example, after presenting Theorem 3.1, a note should be added indicating that its proof is provided in Appendix A.***
>
> As suggested, we have added the necessary references of Appendices in the main text. For example, the proof of Theorem 3.1 is provided in Appendix A; the introduction of datasets is provided in Appendix B.1; more experimental analysis are provided in Appendix B.2.
>
> ***Question 1: What is the form of the function $f (\\cdot)$? Does it use only anchor vectors as classifier weights?***
>
> The function $f (\\cdot)$ in the paper is a multivariate function that maps the feature vector $\\boldsymbol{x}$ to the label distribution. Specifically, it uses a softmax normalization over the dot products between the feature vector $\\boldsymbol{x}$ and the anchor vectors $\\{\\boldsymbol \\omega _ m\\} _ \{m=1\}^M$. The form of $f(\\cdot)$ is $softmax([\\langle \\boldsymbol\\omega_m, \\boldsymbol x\\rangle ] _ \{m=1\}^M)$ where $\\langle \\boldsymbol \\omega _ m, \\boldsymbol x \\rangle$ is the dot product between the anchor vector $\\boldsymbol \\omega _ m$ and the feature vector $\\boldsymbol x$, and $softmax(\\cdot)$ ensures the output is a valid probability distribution.
>
> The function $f(\\cdot)$ does not use only anchor vectors as classifier weights. The function $f(\\cdot)$ is theoretically compatible with various representation learning techniques, serving as a classifier when combined with architectures like convolutional neural networks for image classification or graph convolutional networks for node classification. Specifically, in typical label distribution learning (LDL) models, the output can be formulated as $softmax([\\langle \\boldsymbol\\omega _ m, \\boldsymbol v \\rangle ] _ \{m=1\}^M)$, where $\\boldsymbol v$ represents the feature vector of a sample. In deep learning scenarios, $\\boldsymbol v$ is usually derived by passing the raw feature vector $\\boldsymbol x$ through a feature extraction network, whereas in non-deep learning settings, $\\boldsymbol v$ is often directly set to $\\boldsymbol x$. However, in our experiments, we deliberately abstain from incorporating representation learning techniques and instead employ anchor vectors exclusively as classifier weights. This choice is motivated by two key considerations: (1) Most available label distribution datasets are tabular in nature, rendering complex neural network architectures unnecessary. (2) To ensure a fair comparison with state-of-the-art baseline methods, which predominantly do not adopt representation learning techniques, we also avoid a representation learning technique. Thus, our practical implementation of $f(\\cdot)$ in experiments relies solely on anchor vectors as classifier weights.
>
> ***Question 2: Are the anchor vectors treated as learnable parameters? It would be helpful to clarify the background of LDL.***
>
> Yes, the anchor vectors are indeed treated as learnable parameters that play a fundamental role in the model's architecture. These vectors are optimized during minimizing the Kullback-Leibler (KL) divergence between the predicted label distributions and the ground-truth distributions, as formalized in Equation (3) of the paper. The learning process is further refined through the proposed Inter-Anchor Angular Regularization (IAR) term, which explicitly penalizes excessive cohesion between anchor vectors by minimizing their pairwise cosine similarities.
>
> LDL (Label Distribution Learning) is an effective approach to accurately estimate the entire conditional distribution of labels according to a set of feature variables. This task receives increasing attention both in the field of statistics and machine learning, as the information about the entire distribution is crucial in scenarios that are sensitive to risk, extremes, or uncertainty, such as drug efficacy prediction or emotion recognition. Various kinds of techniques, such as model calibration or mixture density neural network can be utilized to estimate the entire conditional distribution using the training samples that are labeled only with the mean or mode of the underlying true conditional distribution, which are beneficial in the tasks where the true label distributions are unavailable. However, there remain a large number of real-world scenarios where the true distributions are readily available. To address this kind of scenarios, LDL is proposed to learn a multivariate regressor that maps a set of feature variables to the entire conditional distribution of labels according to a training set where each instance is labeled with a label distribution. Compared to the cases without true label distributions, LDL is capable of predicting the entire conditional distribution of labels more accurately, as it is directly supervised by the true label distributions.

---

> > ### Comment · Reviewer_zUMv · 2025-08-03
> >
> > Thank you for your rebuttal. All of my concerns have been addressed, and I am inclined to maintain a positive score for this work.

---

### Official Review · Reviewer_r9iZ · 2025-06-28

**Clarity:** 3
**Significance:** 2
**Originality:** 2
**Rating:** 4
**Confidence:** 2

**Summary:**

The authors demonstrate that many Label Distribution Learning (LDL) methods tend to make significantly larger errors on low-entropy instances compared to high-entropy ones, a disparity that standard aggregate metrics often obscure. To address this, they present Theorem 3.1, which links the cohesion—measured by small pairwise angles—of anchor vectors in softmax models to a lower bound on the entropy of the predicted label distributions. This theoretical insight helps explain why low-entropy samples are particularly challenging to model accurately. To mitigate this issue, the authors introduce an Inter-Anchor Angular Regularizer (IAR) that penalizes overly small angles between anchor vectors, encouraging more dispersed and expressive representations. Additionally, they propose a new evaluation metric, Entropy-Calibrated Aggregation (ECA), which separately averages performance on low- and high-entropy test subsets to provide a more nuanced assessment. Experimental results across eight benchmark datasets using four recent LDL baselines plus ridge regression, along with ablation studies (“ER”, “EW”) and a hyperparameter analysis, show that their approach yields substantial improvements on low-entropy samples without degrading performance on high-entropy ones. Overall, their method achieves the best ECA scores on six out of eight datasets.

**Questions:**

See W1

**Ethical Concerns:**

["NO or VERY MINOR ethics concerns only"]

**Limitations:**

yes

**Quality:**

3

**Strengths And Weaknesses:**

Strengths:
1. Connecting anchor geometry to entropy bias is simple yet convincing; the angle-entropy bound is intuitive and mathematically clean.
2. Clear assumptions and and full proofs

Weaknesses:
1. All baselines are non-deep or shallow deep models. Demonstrating IAR on a modern backbone (e.g., ViT-LDL) would strengthen the claim of universality.

---

> ### Author Rebuttal · Authors · 2025-07-30
>
> Dear Reviewer r9iZ,
>
> Thank you for your positive assessment on our work. We sincerely appreciate the time and effort that you have dedicated to evaluating our work. We have carefully considered all the comments and have revised the paper accordingly. Below, we provide a point-by-point response to your suggestions.
>
> ***Question: All baselines are non-deep or shallow deep models. Demonstrating IAR on a modern backbone (e.g., ViT-LDL) would strengthen the claim of universality.***
>
> We selected two datasets where instance features are encoded by raw image information to demonstrate the effectiveness of the proposed IAR term in deep learning models. We apply the proposed IAR term to the ViT-LDL algorithm while maintaining strict consistency with the experimental procedure utilized in the main text. The next two tables show the performance of the ViT-LDL model with IAR term and the ViT-LDL model without IAR term. It can be seen from the next two tables that our proposed IAR improves the performance of ViT-LDL in most cases.
>
> |   Jaffe  | ViT-LDL with IAR  | ViT-LDL without IAR |
> | :------------- | :---------------- | :------------------ |
> | KL (LEA)       | 0.024$\\pm$0.0159  | 0.0329$\\pm$0.0167   |
> | KL (HEA)       | 0.0153$\\pm$0.003  | 0.0154$\\pm$0.003    |
> | KL (ECA)       | 0.02$\\pm$0.0105   | 0.0244$\\pm$0.0117   |
> | Cosine (LEA)   | 0.9784$\\pm$0.0148 | 0.9699$\\pm$0.0171   |
> | Cosine (HEA)   | 0.9851$\\pm$0.0029 | 0.985$\\pm$0.0032    |
> | Cosine (ECA)   | 0.9817$\\pm$0.0095 | 0.9774$\\pm$0.0113   |
> | Cheb (LEA)     | 0.0719$\\pm$0.0213 | 0.09$\\pm$0.0251     |
> | Cheb (HEA)     | 0.0474$\\pm$0.0039 | 0.0481$\\pm$0.0047   |
> | Cheb (ECA)     | 0.0597$\\pm$0.015  | 0.0693$\\pm$0.0184   |
> | Intersec (LEA) | 0.921$\\pm$0.0219  | 0.9019$\\pm$0.0233   |
> | Intersec (HEA) | 0.935$\\pm$0.0048  | 0.9354$\\pm$0.0056   |
> | Intersec (ECA) | 0.9281$\\pm$0.0154 | 0.9187$\\pm$0.0175   |
>
> |     Emotion6          |ViT-LDL with IAR  | ViT-LDL without IAR |
> |:--------------|:------------------|:------------------|
> | KL (LEA)        | 0.1094$\\pm$0.032  | 0.1091$\\pm$0.0267 |
> | KL (HEA)       | 0.1461$\\pm$0.0533 | 0.1327$\\pm$0.0307 |
> | KL (ECA)        | 0.1372$\\pm$0.0379 | 0.129$\\pm$0.0234  |
> | Cosine (LEA)    | 0.9712$\\pm$0.0038 | 0.9652$\\pm$0.0068 |
> | Cosine (HEA)   | 0.9724$\\pm$0.0056 | 0.9721$\\pm$0.0048 |
> | Cosine (ECA)    | 0.9677$\\pm$0.0065 | 0.966$\\pm$0.0048  |
> | Cheb (LEA)      | 0.1118$\\pm$0.014  | 0.1239$\\pm$0.0163 |
> | Cheb (HEA)     | 0.119$\\pm$0.0197  | 0.1202$\\pm$0.0174 |
> | Cheb (ECA)      | 0.1235$\\pm$0.0156 | 0.1284$\\pm$0.0118 |
> | Intersec (LEA)  | 0.8771$\\pm$0.011  | 0.8652$\\pm$0.0137 |
> | Intersec (HEA) | 0.8675$\\pm$0.0166 | 0.8665$\\pm$0.0151 |
> | Intersec (ECA)  | 0.8635$\\pm$0.0132 | 0.8596$\\pm$0.0098 |

---

> > ### Comment · Reviewer_r9iZ · 2025-08-05
> >
> > I appreciate the authors for providing the experimental results on ViT backbones, which appear very promising. As I am not an expert in the LDL domain, I will refrain from changing my rating. Nonetheless, based on my current understanding and the presented empirical results, I find this paper to be compelling.

---

### Official Review · Reviewer_21xv · 2025-06-29

**Clarity:** 3
**Significance:** 3
**Originality:** 3
**Rating:** 5
**Confidence:** 5

**Summary:**

This paper identifies and addresses the "entropy bias" in Label Distribution Learning (LDL), where existing algorithms perform poorly on low-entropy samples, which are crucial for decision-making. The authors propose two key innovations:

- Inter-anchor Angular Regularization (IAR): A regularization term that penalizes overly cohesive anchor vectors to better capture low-entropy distributions.

- Entropy-Calibrated Aggregation (ECA): An evaluation strategy that balances performance assessment across low- and high-entropy test samples.

The paper offers both theoretical analysis (including an entropy lower bound theorem) and extensive experiments on eight datasets, demonstrating consistent performance gains, especially on low-entropy samples.

**Questions:**

1. Can IAR be extended to architectures without explicit anchor vectors, such as attention-based or Transformer models?
2. How sensitive is the performance to the choice of entropy threshold in ECA? Would a learned threshold work better?

**Ethical Concerns:**

["NO or VERY MINOR ethics concerns only"]

**Final Justification:**

All my concerns have been resolved, and I am now inclined to recommend acceptance of the paper.

**Limitations:**

yes

**Paper Formatting Concerns:**

The paper formatting appears to meet conference standards with no major concerns.

**Quality:**

3

**Strengths And Weaknesses:**

Strengths:

1. Novel identification of entropy bias in LDL, a previously overlooked but practically important issue.
2. Strong theoretical analysis supporting the IAR formulation, with well-structured theorems and proofs.
3. Comprehensive experiments, including multiple baselines, evaluation metrics, ablations, and significance testing.

Weaknesses:

1. The ECA evaluation section is mathematically overloaded and may benefit from simplification or partial relocation to the appendix.
2. There is limited discussion on applicability to non-anchor-based architectures such as Transformer-based LDL models.

---

> ### Author Rebuttal · Authors · 2025-07-30
>
> Dear Reviewer 21xv,
>
> Thank you for your positive assessment on our work. We sincerely appreciate the time and effort that you have dedicated to evaluating our work. We have carefully considered all the comments and have revised the paper accordingly. Below, we provide a point-by-point response to your suggestions.
>
> ***Question 1: Can IAR be extended to architectures without explicit anchor vectors, such as attention-based or Transformer models?***
>
> Our current theoretical framework and methodology are specifically designed for models with explicit anchor vectors, as the core mechanism relies on regularizing the angular separation between these anchor vectors. Therefore, our method is not applicable to models that do not involve anchor vectors. As elaborated in Section 6 of the main text, our proposed inter-anchor angular regularization (IAR) term is not directly compatible with tree-based LDL algorithms since their learning process does not incorporate anchor vectors. On the other hand, our approach actually can be integrated with transformer or attention-based architectures, as the IAR term solely constrains the weight matrix of the neural network's output layer.
>
> ***Question 2: How sensitive is the performance to the choice of entropy threshold in ECA? Would a learned threshold work better?***
>
> Actually, the entropy threshold does not participate in the model learning process; rather, it serves as a parameter in the model evaluation phase. The choice of this threshold parameter can indeed significantly impact the presented performance. For instance, if the threshold is set too small, the majority of test samples would be treated as high-entropy samples, thereby biasing the evaluation results toward high-entropy samples. To address this problem, our paper proposes treating the threshold as a distribution and computing the mean performance across all possible threshold values. In practice, a more reasonable approach is to determine the threshold based on specific task requirements—that is, defining what entropy level qualifies a sample as high-entropy or low-entropy according to the application's needs. As for whether a learnable threshold could be better, we argue that adaptive threshold would not be particularly beneficial, since the threshold is purely a parameter in model evaluation and does not participate in model training.

---

> > ### Comment · Reviewer_21xv · 2025-08-06
> >
> > Thank you for your response. My concerns have been addressed, and I will raise my score accordingly.

---

### Official Review · Reviewer_3R2w · 2025-07-01

**Clarity:** 2
**Significance:** 2
**Originality:** 3
**Rating:** 4
**Confidence:** 4

**Summary:**

This paper finds that excessive cohesion between anchor vectors contributes significantly to the observed entropy bias phenomenon in LDL algorithms. It accordingly proposes an inter-anchor angular regularization term that mitigates cohesion among anchor vectors by penalizing over-small angles. To alleviate the numerical imbalance of high-entropy samples in the test set, it proposes an entropy-calibrated aggregation strategy that obtains the overall model performance by evaluating performance on the low-entropy and high-entropy subsets of the overall test set separately. Experiments demonstrate the effectiveness of the proposed method.  The main contributions of this paper are:

- It analyzes the generation mechanism of entropy bias from both empirical and theoretical perspectives, and consequently proposes an assumption that the underperformance of LDL models on low-entropy samples is significantly driven by the cohesion of anchor vectors.

- It proposes IAR (i.e., an Inter-anchor Angular Regularization term) to penalize the anchor vectors with over-small angles.

- It proposes ECA (i.e., an Entropy-Calibrated Aggregation strategy) to calculate the overall model performance.

**Questions:**

1. How about the performances of different algorithms on other LDL metrics (e.g., Chebyshev distance, Intersection similarity, etc.)?

2. On some high-entropy datasets, the performance gap between IAR and other algorithms is not significant, while on low-entropy datasets, IAR shows a noticeable performance gap compared to some SOTA algorithms. The authors should further analyze the underlying reasons for these phenomena.

3. The impact of the hyperparameter $\alpha$ on model performance varies across different datasets. The authors should further analyze the underlying reasons for the hyperparameter $\alpha$ on model performance varies across different datasets, and provide practical recommendations for its application.

**Ethical Concerns:**

["NO or VERY MINOR ethics concerns only"]

**Final Justification:**

After reading the reviews' comments and the authors' responses, I would like to give a positive rating for this paper.

**Limitations:**

The authors could further discuss the potential societal impact of the proposed method.

**Quality:**

3

**Strengths And Weaknesses:**

**Strengths**

- The problem studied in this paper is interesting and valuable.
- The paper is well-organized, which is easy to follow.
- The theoretical work improves the value of the paper.



**Weakness**

- The paper primarily uses KL divergence and cosine similarity as evaluation metrics. Although these two metrics can measure the differences between the model outputs and the true label distribution from different perspectives, they may not fully reflect the model's performance in real-world applications. For instance, in tasks that require accurate prediction confidence or uncertainty estimation, relying solely on KL divergence and cosine similarity may be insufficient to thoroughly assess the model’s reliability and effectiveness.

- Although the paper demonstrates the stability of the results through multiple randomized experiments and reports the mean and standard deviation, the analysis of the statistical significance of performance differences across different datasets and algorithms is not sufficiently thorough. For example, on some high-entropy datasets, the performance gap between IAR and other algorithms is not significant, while on low-entropy datasets, IAR shows a noticeable performance gap compared to some SOTA algorithms. These issues are not adequately explored or explained. The authors should further analyze the underlying reasons for these phenomena.
- The impact of the hyperparameter $\alpha$ on model performance varies across different datasets. The authors should further analyze the underlying reasons for this variation and provide practical recommendations for its application.

---

> ### Author Rebuttal · Authors · 2025-07-29
>
> Dear Reviewer 3R2w,
>
> Thank you for your constructive comments on our paper. We sincerely appreciate the time and effort that you have dedicated to evaluating our work. We have carefully considered all the comments and have revised the paper accordingly. Below, we provide a point-by-point response to your suggestions.
>
> ***Question 1: How about the performances of different algorithms on other LDL metrics (e.g., Chebyshev distance, Intersection similarity, etc.)?***
>
> Due to space limitations, we have only shown the performance of the algorithm using KL divergence and cosine similarity measures. Nevertheless, we have shown the performance of the algorithm using Chebyshev distance and intersection similarity in the table below. Due to the character limit for the rebuttal, we only present results under the ECA aggregation strategy for these two metrics, omitting detailed comparisons between low- and high-entropy subsets.
>
> |  Algorithm | ECA (Chebyshev Distance)         | ECA (Intersection Similarity)     | Dataset           |
> |:------------|:--------------------|:--------------------|:------------------|
> | IAR      | (1) 0.082$\pm$0.005 | (1) 0.894$\pm$0.005 | Jaffe            |
> | LDM      | (4) 0.099$\pm$0.012 | (4) 0.881$\pm$0.012 | Jaffe            |
> | DPA      | (6) 0.115$\pm$0.012 | (6) 0.852$\pm$0.013 | Jaffe            |
> | FCC      | (3) 0.092$\pm$0.007 | (3) 0.888$\pm$0.006 | Jaffe            |
> | LRR      | (2) 0.092$\pm$0.007 | (2) 0.889$\pm$0.006 | Jaffe            |
> | Ridge        | (5) 0.115$\pm$0.014 | (5) 0.853$\pm$0.015 | Jaffe            |
> | IAR      | (1) 0.108$\pm$0.003 | (1) 0.871$\pm$0.003 | BU-3DFE          |
> | LDM      | (6) 0.119$\pm$0.007 | (6) 0.863$\pm$0.007 | BU-3DFE          |
> | DPA      | (2) 0.109$\pm$0.003 | (2) 0.870$\pm$0.003 |  BU-3DFE          |
> | FCC      | (5) 0.116$\pm$0.003 | (5) 0.865$\pm$0.003 | BU-3DFE          |
> | LRR      | (4) 0.115$\pm$0.005 | (4) 0.866$\pm$0.004 | BU-3DFE          |
> | Ridge        | (3) 0.109$\pm$0.003 | (3) 0.870$\pm$0.003 | BU-3DFE          |
> | IAR      | (1) 0.313$\pm$0.006 | (1) 0.577$\pm$0.006 | Natural Scene     |
> | LDM      | (6) 0.362$\pm$0.012 | (6) 0.538$\pm$0.010 | Natural Scene     |
> | DPA      | (3) 0.325$\pm$0.010 | (3) 0.571$\pm$0.009 | Natural Scene     |
> | FCC      | (5) 0.358$\pm$0.009 | (5) 0.542$\pm$0.009 | Natural Scene     |
> | LRR      | (4) 0.336$\pm$0.010 | (4) 0.563$\pm$0.010 | Natural Scene     |
> | Ridge        | (2) 0.322$\pm$0.008 | (2) 0.574$\pm$0.008 | Natural Scene     |
> | IAR      | (1) 0.357$\pm$0.016 | (1) 0.555$\pm$0.014 | Emotion6          |
> | LDM      | (6) 0.371$\pm$0.014 | (6) 0.544$\pm$0.012 | Emotion6          |
> | DPA      | (3) 0.360$\pm$0.016 | (2) 0.552$\pm$0.013 | Emotion6          |
> | FCC      | (5) 0.365$\pm$0.013 | (4) 0.548$\pm$0.012 | Emotion6          |
> | LRR      | (4) 0.365$\pm$0.013 | (5) 0.548$\pm$0.012 | Emotion6          |
> | Ridge        | (2) 0.360$\pm$0.017 | (3) 0.552$\pm$0.015 | Emotion6          |
> | IAR      | (1) 0.295$\pm$0.049 | (1) 0.581$\pm$0.036 | Art Painting |
> | LDM      | (4) 0.320$\pm$0.040 | (4) 0.557$\pm$0.027 | Art Painting |
> | DPA      | (5) 0.344$\pm$0.047 | (5) 0.529$\pm$0.041 | Art Painting |
> | FCC      | (3) 0.316$\pm$0.043 | (3) 0.562$\pm$0.032 | Art Painting |
> | LRR      | (2) 0.316$\pm$0.043 | (2) 0.563$\pm$0.032 | Art Painting |
> | Ridge        | (6) 0.345$\pm$0.046 | (6) 0.528$\pm$0.039 | Art Painting |
> | IAR      | (4) 0.078$\pm$0.003 | (4) 0.802$\pm$0.008 | Music Mood        |
> | LDM      | (3) 0.077$\pm$0.004 | (3) 0.803$\pm$0.008 | Music Mood        |
> | DPA      | (5) 0.085$\pm$0.003 | (5) 0.789$\pm$0.010 | Music Mood        |
> | FCC      | (1) 0.076$\pm$0.003 | (1) 0.803$\pm$0.007 | Music Mood        |
> | LRR      | (2) 0.076$\pm$0.003 | (2) 0.803$\pm$0.007 | Music Mood        |
> | Ridge        | (6) 0.086$\pm$0.004 | (6) 0.789$\pm$0.009 | Music Mood        |
> | IAR      | (1) 0.409$\pm$0.011 | (1) 0.584$\pm$0.010 | M2B               |
> | LDM      | (3) 0.421$\pm$0.020 | (3) 0.571$\pm$0.021 | M2B               |
> | DPA      | (5) 0.430$\pm$0.020 | (5) 0.563$\pm$0.019 | M2B               |
> | FCC      | (2) 0.419$\pm$0.016 | (2) 0.574$\pm$0.016 | M2B               |
> | LRR      | (4) 0.423$\pm$0.018 | (4) 0.571$\pm$0.018 | M2B               |
> | Ridge        | (6) 0.431$\pm$0.020 | (6) 0.562$\pm$0.020 | M2B               |
> | IAR      | (1) 0.210$\pm$0.024 | (1) 0.749$\pm$0.022 | Movie             |
> | LDM      | (4) 0.213$\pm$0.023 | (4) 0.747$\pm$0.021 | Movie             |
> | DPA      | (6) 0.213$\pm$0.025 | (6) 0.746$\pm$0.023 | Movie             |
> | FCC      | (3) 0.212$\pm$0.024 | (3) 0.748$\pm$0.022 | Movie             |
> | LRR      | (2) 0.212$\pm$0.024 | (2) 0.748$\pm$0.022 | Movie             |
> | Ridge        | (5) 0.213$\pm$0.024 | (5) 0.746$\pm$0.023 | Movie
>
> According to the results in the above table, it can be seen that under the Chebyshev distance and intersection similarity metrics, our algorithm achieves state-of-the-art performance on the Jaffe, BU-3DFE, Natural Scene, Emotion6, Abstract Painting, M2B, and Movie datasets. Notably, while it does not attain the best performance on the Music Mood dataset, it remains highly competitive, with performance very close to the top.
>
> ***Question 2: On some high-entropy datasets, the performance gap between IAR and other algorithms is not significant, while on low-entropy datasets, IAR shows a noticeable performance gap compared to some SOTA algorithms. The authors should further analyze the underlying reasons for these phenomena.***
>
> The performance gap is more pronounced on low-entropy datasets than on high-entropy datasets because prediction errors exhibit greater variance in low-entropy datasets compared to high-entropy ones. We demonstrate this claim through data simulation with the following procedure:
>
> 1. First, we uniformly generate 10,000 label distributions $[\boldsymbol z_n]_{n=1}^{10,000}$ with varying entropy levels.
> 2. For each label distribution $\boldsymbol{z}_n$, we randomly generate 10,000 predicted label distributions and compute their KL divergence, cosine similarity, Chebyshev distance, and intersection similarity with respect to $\boldsymbol{z}_n$.
> 3. We then partition $[\boldsymbol z_n]_{n=1}^{10,000}$ into nine groups based on entropy intervals: [0, 0.2], (0.2, 0.4], (0.4, 0.6], (0.6, 0.8], (0.8, 1], (1, 1.2], (1.2, 1.4], (1.4, 1.6], (1.6, $+\infty$).
> 4. Finally, we compute the variance of all prediction performance metrics within each group.
>
> The results (shown in the table below) reveal that label distributions with higher entropy exhibit smaller variance in prediction errors. This explains why the performance gap is more noticeable in low-entropy datasets than in high-entropy ones.
>
> | Entropy Range    | KL Variance | Cosine Variance | Chebyshev Variance | Intersection Variance |
> | ---------------- | ----------- | --------------- | ------------------ | --------------------- |
> | [0, 0.2]         | 0.8555      | 0.0453          | 0.0129             | 0.0158                |
> | (0.2, 0.4]       | 0.6871      | 0.0436          | 0.0128             | 0.0154                |
> | (0.4, 0.6]       | 0.6011      | 0.0411          | 0.0127             | 0.0152                |
> | (0.6, 0.8]       | 0.4107      | 0.0376          | 0.0102             | 0.0149                |
> | (0.8, 1]         | 0.2854      | 0.0321          | 0.0101             | 0.0134                |
> | (1, 1.2]         | 0.232       | 0.0285          | 0.0098             | 0.0128                |
> | (1.2, 1.4]       | 0.1486      | 0.0215          | 0.0069             | 0.0126                |
> | (1.4, 1.6]       | 0.0941      | 0.0115          | 0.0054             | 0.0099                |
> | (1.6, $+\infty$) | 0.0459      | 0.0044          | 0.0042             | 0.0081                |
>
> ***Question 3: The impact of the hyperparameter $\alpha$ on model performance varies across different datasets. The authors should further analyze the underlying reasons for the hyperparameter $\alpha$ on model performance varies across different datasets, and provide practical recommendations for its application.***
>
> We analyzed the underlying reasons for the hyperparameter $\alpha$ on model performance varies across different datasets, and provided practical recommendations for its application as follows.
>
> **Underlying reasons for the hyperparameter $\alpha$ on model performance varies across different datasets.** The hyperparameter $\alpha$ balances the relative importance between the training error (KL divergence between the model-output label distribution and the true label distribution) and the IAR regularization. Since the converged training error (KL divergence value) is influenced by the number of labels, data distribution, and the noise level of the dataset, the optimal weight ($\alpha$) for the IAR term should be dataset-dependent. Furthermore, variations in data distribution across datasets lead to different degrees of over-uniformity when the training error is converged, consequently requiring different levels of IAR regularization during training. This further necessitates dataset-specific $\alpha$ values.
>
> **Recommended settings.** As analyzed in lines 249-254 of our paper, we recommend $\alpha=40$ as a default setting for most datasets, as it approaches optimal performance across diverse scenarios. For further performance optimization, we suggest the following empirical guidelines:
>
> 1. When the training set contains predominantly high-entropy label distributions, consider using $\alpha<40$.
> 2. For datasets with predominantly low-entropy label distributions, larger $\alpha$ values are preferable.

---

> ### Comment · Reviewer_3R2w · 2025-08-05
>
> Thanks for the responses. I appreciate the further results on more evaluation metrics and the analysis of the impact of the hyperparameter $\alpha$. However, for Q2, although the authors give a simulation-based explanation of how prediction error variance differs across entropy levels, I believe the paper would benefit from the analysis of actual data or a theoretical perspective to further study the phenomenon.

---

> ### Author Response · Authors · 2025-08-06
>
> We sincerely appreciate the reviewer's insightful comments regarding the need for further validation on real-world data and theoretical analysis. In response, we have conducted additional experiments and analyses as follows:
>
> **​​Real-world Data Validation**: ​​We have verified this phenomenon on real-world datasets by replacing the artificially generated label distributions in our simulation experiments with actual dataset label distributions. To facilitate comparison across datasets with varying dimensions of label distributions, we present normalized entropy values rather than raw entropy values in the following table (as different datasets have different dimensionality in their label distributions, making raw entropy values incomparable).
> The experimental results on real-world data consistently demonstrate that high-entropy datasets (Jaffe, BU-3DFE, Movie, Music Mood) exhibit significantly lower variance in prediction performance compared to low-entropy datasets (Natural Scene, Emotion6, Art Painting, M2B).
>
> | Cheb  Variance |KL Variance |Cosine Variance| Intersec  Variance | Dataset           | Entropy Range   | Entropy Distribution    |   # Labels |
> |-----------:|-----------:|----------:|-----------:|:------------------|:----------------|:------------------------|-----------:|
> | 0.00503105 |  0.0226465 | 0.0110323 | 0.00882119 | Jaffe            | [0.862, 0.998]  | 0.959$\\pm$0.026        |          6 |
> | 0.00634856 |  0.0238445 | 0.013278  | 0.00938306 | BU-3DFE          | [0.841, 1.000]  | 0.953$\\pm$0.037        |          6 |
> | 0.00812684 |  0.130914  | 0.0201881 | 0.0142667  | Movie             | [0.422, 0.999]  | 0.878$\\pm$0.061        |          5 |
> | 0.0015705  |  0.0346131 | 0.0092632 | 0.00743669 | Music Mood        | [0.816, 0.997]  | 0.944$\\pm$0.034        |          9 |
> | 0.0732037  |  1.69887   | 0.0385863 | 0.0349031  | Natural Scene     | [0.006, 0.941]  | 0.466$\\pm$0.272        |          9 |
> | 0.0224852  | 12.7154    | 0.0319004 | 0.0193312  | Emotion6          | [0.126, 0.976]  | 0.640$\\pm$0.156        |          7 |
> | 0.0126449  | 10.8347    | 0.0255178 | 0.0161054  | Art Painting | [0.124, 0.965]  | 0.716$\\pm$0.126        |          8 |
> | 0.0263162  |  0.431754  | 0.041684  | 0.0187049  | M2B               | [0.066, 0.692]  | 0.407$\\pm$0.118        |          5 |
>
> We also note an interesting observation within the low-entropy dataset group: while Emotion6 and Art Painting show similar entropy levels to other low-entropy datasets, their KL divergence variance is substantially higher. We believe that this occurs because the KL metric is particularly sensitive to zero elements, and these two datasets contain abundant zero elements in their label distributions.
>
> **Theoretical Validation**: Given a true label distribution $[d_1,d_2,\dots,d_n]$, where $d_1<d_2<\cdots<d_n$ can be assumed without loss of generality, the squared Euclidean distance between any predicted label distribution $[z_1,z_2,\dots,z_n]$ and the true label distribution is $y=(d_1-z_1)^2+(d_2-z_2)^2+\cdots+(d_n-z_n)^2$, which can be utilized to quantify the performance the label distribution predictor. The minimum value of $y$ is zero when $z_i=d_i$ holds for $i=1,2,\dots,n$. The maximum value of $y$ is $(1-d_1)^2+d_2^2+\cdots+d_n^2$ when $z_1=1$, $z_2=z_3=\cdots=z_n=0$. Then the range of the performance is $r=(1-d_1)^2+d_2^2+\cdots+d_n^2$.
> Now, let $\boldsymbol d = [d_1,d_2,\dots,d_n]$ and $\boldsymbol h = [h_1,h_2,\dots,h_n]$ denote two true label distributions, where the Gini coefficient of $\boldsymbol d$ is smaller than the Gini coefficient of $\boldsymbol h$, i.e., $(1-h_1^2-\cdots-h_n^2) - (1-d_1^2-\cdots-d_n^2) = c > 0$, and $h_1+c/2>d_1$. It should be noted that we use Gini coefficient here to quantify the uncertainty of a label distribution, which is similar to entropy yet easier to calculate than entropy. According to $(1-h_1^2-\cdots-h_n^2) - (1-d_1^2-\cdots-d_n^2) = c > 0$, we have:
>
> $(1-h_1^2-\cdots-h_n^2) - (1-d_1^2-\cdots-d_n^2) = c > 0$
>
> $\Longleftrightarrow d_1^2+d_2^2+\cdots+d_n^2 =  h_1^2+h_2^2+\cdots+h_n^2 + c > h_1^2+h_2^2+\cdots+h_n^2 + 2(d_1 - h_1)$
>
> $\Longleftrightarrow d_1^2+d_2^2+\cdots+d_n^2 - 2d_1+1 >  h_1^2+h_2^2+\cdots+h_n^2 - 2h_1+1$
>
> $\Longleftrightarrow (1-d_1)^2+d_2^2+\cdots+d_n^2 > (1-h_1)^2+h_2^2+\cdots+h_n^2$
>
> Therefore, the performance range on a true label distribution with lower uncertainty is broader than the performance range on a true label distribution with higher uncertainty if $h_1+c/2>d_1$ (this requirement can be satisfied in most practical cases (about 90% cases in the real-world datasets in our paper)).
>
> We believe these additional experiments and theoretical analyses substantially strengthen our original findings and provide more comprehensive evidence for the observed phenomenon. Please don’t hesitate to let us know if additional details or revisions would be helpful if there are any remaining points that require further clarification.

---

> > ### Comment · Reviewer_3R2w · 2025-08-07
> >
> > Thanks for the responses. I have no further questions.

---

### Decision · Program_Chairs · 2025-09-17

**Decision:**

Accept (poster)

**Comment:**

This paper identifies and analyzes entropy bias in Label Distribution Learning, where models struggle with low-entropy samples. The authors propose an inter-anchor angular regularization term and an entropy-calibrated aggregation strategy, supported by both theory and extensive experiments. Reviewers found the theoretical analysis clear and the empirical validation convincing, with rebuttal additions (new metrics, ViT results, real-data variance analysis) further strengthening the work. While some issues remain around hyperparameter sensitivity and broader applicability, the paper makes a solid contribution.